# Undercurrents in the Northeastern Black Sea Detected on the Basis of Multi-Model Experiments and Observations

**Sergey G. Demyshev** [1], **Olga A. Dymova** [1], **Natalia V. Markova** [1], **Evgenia A. Korshenko** [2],
**Maksim V. Senderov** [1], **Nikita A. Turko** [3] **and Konstantin V. Ushakov** [4,3,*]

1   Marine Hydrophysical Institute, Russian Academy of Sciences, Kapitanskaya Str. 2,
    299011 Sevastopol, Russia; demyshev@gmail.com (S.G.D.); olgdymova@rambler.ru (O.A.D.);
    n.v.markova@mail.ru (N.V.M.); senderovmaxim@gmail.com (M.V.S.)
2   N.N. Zubov State Oceanographic Institute, Roshydromet, Kropotkinskiy Lane 6, 119034 Moscow, Russia;
    kekapon@gmail.com
3   Moscow Institute of Physics and Technology, Institutskiy Per. 9, 141701 Dolgoprudny, Russia;
    turko@phystech.edu
4   Shirshov Institute of Oceanology, Russian Academy of Sciences, Nahimovskiy Prospekt 36,
    117997 Moscow, Russia
*   Correspondence: ushakovkv@mail.ru

**Abstract:** Numerical simulation results of the Black Sea circulation obtained by four ocean dynamics models are compared to each other and to in situ data in order to determine the features of the Black Sea deep-water circulation such as deep-water undercurrents. The year 2011 is chosen as the test period due to the availability of deep-sea observations, including ARGO profiles and ADCP current velocities. Validation of the simulation results is based on comparison with the temperature and salinity measured by the ARGO floats. Anticyclonic currents (undercurrents) under the cyclonic Rim Current are detected by the results of all numerical models near the North Caucasian coast. The main characteristics of undercurrents are consistent with in situ data on current velocity up to a depth of 1000 m obtained by the Aqualog probe at the IO RAS test site near Gelendzhik in June 2011. The analysis of the spatio-temporal variability of the modeled salinity and velocity fields reveals that the most probable origin of the undercurrents is the horizontal density gradient of seawater in the region.

**Keywords:** Black Sea; undercurrents; numerical modeling; coordinated experiments

## 1. Introduction

Undercurrents are a well-known phenomenon in the World Ocean. These are currents that propagate in the deep layers of the ocean with a direction which is usually opposite to the surface circulation. Examples of such currents are the Cromwell Current in the Pacific Ocean [1], the South Equatorial Undercurrent in the Atlantic Ocean [2], the Agulhas Undercurrent in the Indian Ocean [3], and others. The generation of undercurrents is associated with pressure gradients [4], the configuration of the bottom relief [5] including the JEBAR effect [6], the thermohaline structure of waters, and other factors [7]. Additionally, undercurrents could be formed by mesoscale eddies passing along the continental slope. Such a mechanism was presented for the Bay of Bengal in [8]. Eddies can directly generate currents between their periphery and the continental slope, as well as change the isopycnic surfaces, which can lead to the formation of internal waves and gradient currents.

In the Black Sea, the long-standing question of the presence of a deep anticyclonic current under the Main Black Sea Current (the Rim Current, which encircles the basin along the periphery in the upper sea layer) has not yet been resolved. There is evidence of the presence of anticyclonic circulation elements under the main pycnocline, obtained both on the basis of modeling and observations [9–14]. In the continental slope zone at a depth of 1700 m, deep-water current velocities up to 13 cm/s were recently observed in a

one-year field experiment using a moored ADCP [15]. This experiment also showed that the direction of the measured currents changed on a scale of several weeks.

There are also arguments in favor of the absence of a basin-scale undercurrent flow [16,17]. When studying the dynamics of the Black Sea on the basis of the MHI model [18], anticyclonic jet currents were found in the velocity field, propagating along the continental slope in some areas. In the northeastern part of the sea, the modeled undercurrent was found quasi-periodically, in springs and summers of several years.

The complexity of studying undercurrents is primarily due to the impossibility of performing extensive regular full-scale measurements of the current velocity. On the other hand, the operative system of Argo floats [19] does not give direct information about current velocities but provides temperature and salinity profiles for assimilation in models and for validation of modeling results. Supported by Argo data, numerical modeling remains one of the most complete and cost-effective ways to reconstruct the characteristics of the marine environment in all parts of the World Ocean. It allows one to investigate dynamic, thermohaline, biogeochemical, and other parameters of the ocean state at various spatio-temporal scales with the necessary discreteness, assess the variability of three-dimensional fields, and make forecasts for the near and distant future.

Despite a large number of works on the reconstruction of the Black Sea circulation, the majority of them focus on the study of the processes occurring in the upper 200–300 m layer of the sea. The main goal of our work is to clarify the features of the Black Sea dynamics in the deep-water layers. The progress of numerical model development since the beginning of the 2000s has not significantly affected the problem of studying the system of deep currents of the Black Sea. A relatively small amount (and often low quality) of observational data accumulated over the entire period of deep-water measurements in the Black Sea does not allow for studying the field of currents only on the basis of the measurement data (e.g., [20]). It also makes it extremely difficult to fine-tune the parameters of numerical models for the water layer under the main pycnocline. At the same time, at horizons deeper than 300 m, the vertical density gradient is small, so even small anomalies in the density field can lead to a significant transformation of the currents. Therefore, in all studies that are based on a single numerical model, biases in deep-water velocity fields are probable. Thus, in this work, the results of four similar numerical experiments carried out with different models of ocean dynamics are compared in order to refine the characteristics of the Black Sea deep-water circulation. The modeling results are assessed by comparing them with the data of deep-water field observations of temperature and salinity (T&S) provided by the ARGO project.

## 2. Data and Methods

At the preliminary stage of the study, a vast amount of available observation data containing measurements in the deep-water regions of the Black Sea was collected and analyzed. The information of contact measurements accumulated in the MHI Bank of Oceanographic Data [21] and the data of the ARGO project were evaluated. Some results will be presented in Section 3.3. Considering that the simulation results should be compared with the most complete and accurate set of in situ data, the time interval from January to December 2011 was chosen for carrying out multi-model numerical experiments. For this period, ARGO data on deep-water T&S profiles were available, and the data of deep-water current measurements by the Aqualog moored profilograph in June 2011 near Gelendzhik [13] were kindly provided by the authors.

Four modern models of ocean dynamics were used as tools for the numerical analysis of the Black Sea circulation. All models are based on the three-dimensional primitive system of equations of ocean thermohydrodynamics in the Boussinesq and hydrostatic approximations.

## 2.1. MHI Eddy-Resolving Model

The MHI model is developed at the Marine Hydrophysical Institute (MHI) of the Russian Academy of Sciences (RAS). It has long been used for Black Sea circulation forecasting [22]. It has the second order of accuracy along spatial coordinates in the case of constant grid step. The model implements the leapfrog time scheme [23], a semi-implicit representation for pressure, and the TVD schemes for approximation of heat and salt advection [24]. The biharmonic horizontal operator is used in the equations for the transport of momentum, heat, and salt. Vertical turbulent processes are parameterized on the basis of the Mellor-Yamada 2.5 theory [25]. The river runoff and water exchange through straits are explicitly taken into account. The spatial resolution of the model configuration used is 1.64 km in the horizontal coordinates, and 27 z-horizons are considered in vertical.

## 2.2. INMOM Sigma-Coordinate Ocean Model

The INMOM model is developed at the Marchuk Institute of Numerical Mathematics (INM RAS). The specific feature of the INMOM numerical implementation is a modular construction principle, which is based on the method of multicomponent splitting [26]. In the model, the lateral viscosity operator is a combination of the 2nd and 4th order operators [27]. The vertical viscosity is defined according to the [28] parameterization. No-flow bottom and lateral boundary conditions are set for temperature, salinity, and normal velocity, supplemented by the no-friction lateral and quadratic friction bottom conditions for the tangential velocity component. The model nudges surface salinity to climatic data by adding a relaxation surface salt flux. The sea surface temperature (SST) is nudged to the SKIRON data (see below) with the same relaxation parameter as in the case of salinity. In addition, the nudging to climatic values [29] is used for model T&S at depths below 150 m with a relaxation period of 120 days. River runoff is prescribed in the form of pseudo-precipitation concentrated in water areas adjacent to river mouths. The INMOM model for the region of the Black, Azov, and Marmara Seas is implemented with the spatial resolution of 1 km, and 20 sigma levels are considered in vertical.

## 2.3. NEMO Model

The model of the European NEMO Consortium [30] has been set up for the Black Sea configuration [31]. The momentum equations are approximated with an energy and enstrophy conserving scheme. For nonlinear terms in the transport–diffusion equations, the TVD scheme is used. Lateral turbulent exchange is described by a biharmonic operator, and the vertical mixing is defined by the k-$\varepsilon$ parameterization. The computational domain is covered by a regular grid with $(1/24)^\circ \times (1/17)^\circ$ resolution in meridional and zonal directions, which corresponds to about 4.6 km. The scheme of time splitting into barotropic (fast) and baroclinic (slow) modes was used with a kinematic surface condition for the sea level calculation. Time steps for the slow and fast modes are 5 min and 10 s, respectively. The time discretization was carried out using a modified leapfrog scheme. The free-slip condition is imposed on the lateral boundaries for the equations of motion, and nonlinear friction is specified on the bottom. In the equations of heat and salt transport and diffusion at the lateral boundaries and the bottom, the no-flux and zero Laplacian conditions are set. The climatic runoffs of 14 rivers are taken into account. Positions of 35 z-horizons are set using an analytical function proposed by the authors of the model.

## 2.4. INMIO Eddy-Resolving Model

The INMIO model is developed at INM RAS and Shirshov Institute of Oceanology (IO RAS) [32,33]. The momentum equations are approximated according to the leapfrog scheme (for advective terms, pressure gradient, and Coriolis force), the Euler scheme (horizontal diffusion), and the Crank–Nicolson scheme (vertical diffusion). The simulation of barotropic dynamics is performed in an explicit way through the two-dimensional system of shallow water equations approximated with the fast scheme with overlapping stencils [34]. The heat and salt advection is implemented according to the flux corrected

transport scheme [35]. The subgrid horizontal tracer and momentum exchange is parameterized by the biharmonic operator. Vertical mixing is defined by the Munk–Anderson scheme [36] with convective adjustment in the case of unstable stratification. For the basin of the Black and Azov seas, the model configuration was set up on a regular grid with a resolution of $0.018° \times 0.013°$, which corresponds to steps of approximately 1.5 km. The time step is 1 min for the baroclinic mode and 1.5 s for the barotropic one. At the air–sea interface, the non-linear kinematic free surface condition is used. On the rigid lateral boundaries, the condition of zero momentum flux (including the biharmonic flux component) is set, and for the bottom, there is the quadratic friction term. For tracers, the no-flux conditions are set on rigid boundaries, while at river mouths and Bosphorus Strait, the heat, salt, and water exchange is defined explicitly. The vertical discretization includes 51 horizons with a step ranging from 2 m near the surface to 100 m at depths greater than 1 km. The model is implemented for massively parallel computations under control of the Compact Modeling Framework [37].

### 2.5. Experiment and Validation Setup

Some key parameters of the aforementioned models are shown in Table 1.

**Table 1.** Parameters of the numerical models.

| Model | Vertical Axis | Grid Type | Resolution | Vertical Mixing | Horizontal Mixing | Equation of State | Bulk Formulae |
|-------|---------------|-----------|------------|-----------------|-------------------|-------------------|---------------|
| MHI | 27 z-levels | C | 1.6 km | [25] | biharmonic | [38] | SKIRON and [39] |
| INMOM | 20 $\sigma$-levels | C | 1 km | [28] | 2nd and 4th order | [40] | [41] |
| NEMO | 35 z-levels | C | 4.6 km | k-$\varepsilon$ | biharmonic | [42] | [41] |
| INMIO | 51 z-levels | B | 1.5 km | [36] | biharmonic | [43] | [41] |

The atmospheric forcing for the year 2011 was obtained from the SKIRON database [44] with a resolution of $0.1°$ in latitude and longitude. The SKIRON data were interpolated to grid domain for each model. In the MHI model, the surface fluxes of heat and water were taken directly from the SKIRON, while the wind stress components were calculated through the quadratic drag formula of [39]. The INMOM, NEMO, and INMIO models used the CORE bulk formulae [41] for calculation of all fluxes (see Table 1).

The basin bathymetry is defined by means of the EMODnet depth array of (1/8)′ resolution [45]. Only the INMOM model uses the GEBCO ocean bed topography with a spatial resolution of 30′ [46]. Our analysis showed that the GEBCO and EMODnet arrays differed insignificantly at the model horizontal resolution of the order of 1 km. The INMOM and INMIO computational domains include the Azov Sea, while in the MHI and NEMO simulations, it is replaced with a prescribed seasonally varying flow through the Kerch Strait.

The initial fields (sea level, temperature, salinity, and horizontal velocity) on January 1 were obtained from the experiment [11]. Here, the thermohydrodynamical fields were spun up till establishing a steady state of the Black Sea circulation (about 14 years of model run) with assimilation of the seasonally varying multi-year averaged observational data of T&S [47] and with forcing from monthly mean atmospheric data. The procedure of fast geostrophic adjustment was then performed for four days to obtain the mutual agreement of the spun up fields and the SKIRON atmospheric forcing. Finally, numerical experiments with four models were carried out for a period of one year. The models' output consists of daily fields of temperature, salinity, velocity, and sea level in 2011.

Model fields are validated against the T&S data of ARGO floats, which are available till depths of 500–1500 m. We considered 360 profiles of both T&S (floats No 7900465, 7900466, 1901200, 6900803, 6900804, 6900805) performed in all seasons, mainly in the abyssal part

of the Black Sea. Figure 1 shows the locations of the ARGO float stations in 2011, which were used for validation. Since in 2011, only a few deep-sea T&S profiles were obtained outside the ARGO framework (e.g., MHI Cruize No 69 of R/V Professor Vodyanitskii, 2–8 August 2011 [21]), and they were not considered for validation in order not to disturb the homogeneity of the data. The simulation results were interpolated to the observation points, and then deviations of the model values from the measured ones were calculated.

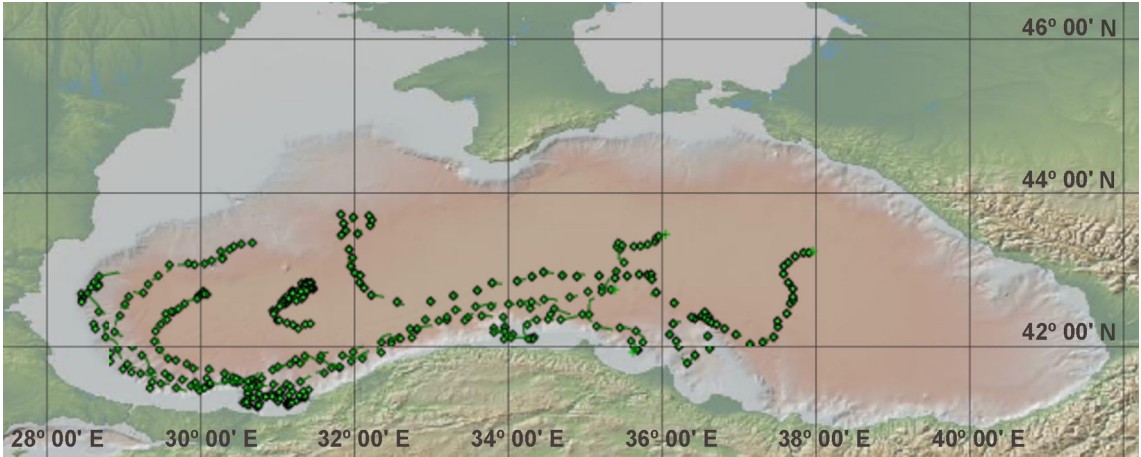

**Figure 1.** Positions of the ARGO float stations with profiling depths of 500–1500 m in 2011.

For further analysis, all the data were grouped according to the sea characteristic layers: the subsurface layer (0–5 m), the upper mixed layer (5–30 m), the cold intermediate layer (CIL, 30–100 m), and the main pycnocline layer (100–300 m). The subpycnocline layer was divided into two parts: the first one from 300 to 800 m and the second one from 800 to 1500 m. The horizon of 800 m was chosen due to the effect of increasing current velocity found in our previous works at depths of 800–1000 m [18,48], which were performed in the framework of MERSEA Class 4 metrics [49]. Finally, 1500 m is the maximum profiling depth of the Black Sea ARGO floats in 2011. Then, at all points of each float's trajectory, series of layer-averaged modeled and measured T&S values were constructed, between which the root-mean-square errors (RMSEs) were calculated. The RMSEs averaged over all trajectories are presented in Table 2.

**Table 2.** Model–observation RMSEs of temperature and salinity for models and depth layers.

| Depth, m | Temperature RMSE, °C | | | | Salinity RMSE, ‰ | | | |
|---|---|---|---|---|---|---|---|---|
| | **MHI** | **INMOM** | **NEMO** | **INMIO** | **MHI** | **INMOM** | **NEMO** | **INMIO** |
| 0–5 | 0.861 | 0.602 | 1.642 | 1.811 | 0.524 | 0.16 | 0.502 | 0.794 |
| 5–30 | 1.815 | 0.436 | 2.499 | 2.971 | 0.205 | 0.149 | 0.440 | 0.436 |
| 30–100 | 0.631 | 0.292 | 0.488 | 2.201 | 0.443 | 0.487 | 0.607 | 0.546 |
| 100–300 | 0.113 | 0.08 | 0.410 | 0.208 | 0.263 | 0.202 | 0.299 | 0.294 |
| 300–800 | 0.047 | 0.052 | 0.017 | 0.039 | 0.087 | 0.067 | 0.085 | 0.084 |
| 800–1500 | 0.03 | 0.125 | 0.013 | 0.109 | 0.009 | 0.008 | 0.009 | 0.013 |

### 3. Comparison with Observational Data

*3.1. Model–Observation Root Mean Square Errors*

The study of T&S fields calculated by the MHI model shows that the largest model–observation differences are observed mainly during the summer period. In the upper mixed layer, the temperature difference reaches several °C in some locations. The highest value of temperature RMSE is equal to 1.82 °C and is found in the 5–30 m layer. The maximal RMSE of salinity is 0.52‰ in the layer 0–5 m. The minimal RMSEs of both T&S are obtained at horizons below 800 m.

The results of the INMOM model show that the maximum of temperature RSME is reached in the subsurface layer and is equal to 0.6 °C. For salinity, the maximum RMSE value refers to the cold intermediate layer and equals 0.5‰. In the subpycnocline layer (300–1500 m), the RMSEs of T&S decrease with depth. However, in the layer 800–1500 m, an increase in RMSE is observed for temperature. The possible reason is that here, the ARGO temperature strongly differs from the monthly mean climatic data, to which the model solution is relaxed below the 150 m horizon. In addition, this may indicate an increased variability in the 800–1500 m layer, leading to intensification of currents.

For the NEMO model, the maximum RMSE for temperature is observed in the 5–30 m layer, and for salinity in the 30–100 m layer, amounting to 2.5 °C and 0.6‰, respectively. The change in temperature RMSE with depth is in qualitative agreement with the MHI data. The minimum deviations are also observed below 800 m.

The INMIO RMSE distribution is qualitatively close to that of the NEMO model. The high temperature deviations in the 0–30 m layers may be caused by the high circulation variability at these depths (including eddy dynamics) and the absence of SST relaxation in both models. The most significant difference is found in a high INMIO RMSE for a temperature of CIL, which is probably due to the too diffusive vertical advection flux corrected scheme of [35].

### 3.2. Sea Circulation Structure

In this section we analyze the structure of the Black Sea circulation simulated by the models. Figures 2 and 3 show the June and December 2011 monthly mean modeled temperature on cross-sections along 43° N and the fields of current velocity at each model's upper horizon.

A brief analysis shows that all models reproduce the main features of the Black Sea circulation. In the current velocity fields, variability of the Rim Current intensity is observed: in summer, the width of the current is greater than in winter, while the velocity is less. The northern part of the gyre is more intense during the warm season. Two synoptic eddies over 100 km in diameter are located near the Crimea and in the southeastern part of the sea—these are the Sevastopol and Batumi anticyclones. In the warm season, the maximum Rim Current velocity is observed in the zone of its interaction with anticyclones. Only in the INMOM model are the synoptic anticyclones at the Rim Current periphery poorly expressed. Synoptic and mesoscale activity differs in four simulations and is associated with different mixing parameterizations and model resolutions.

In the temperature field, all models show the spring–summer warming of the upper sea layers. The CIL (interlayer of cold waters with a temperature of less than 8.35 °C [50]) is located at depths from 40 to 150 m. However, the thickness and continuity of the CIL are different among the model results. For the 30–100 m layer, the smallest RMSE between the modeled and measured temperatures was obtained in the INMOM model data, showing no continuous layer of temperature below 8.35 °C. The heating of subsurface layers according to the MHI and INMIO models is generally less than that according to the NEMO and INMOM.

In winter, all models show a weakening of basin-scale circulation in the western part of the sea and an increase in the eastern one. In monthly mean current fields for December, the Sevastopol anticyclone is poorly developed by MHI and INMOM models. The circulation in the southeastern part of the sea is a system of cyclonic and anticyclonic eddies of varying intensity. As seen from the temperature cross-section, the CIL deepens as a result of winter cooling and intense vertical mixing. The highest temperature in the upper mixed layer reaches 10–11 °C according to the MHI and 13–15 °C in the results of NEMO, INMOM, and INMIO.

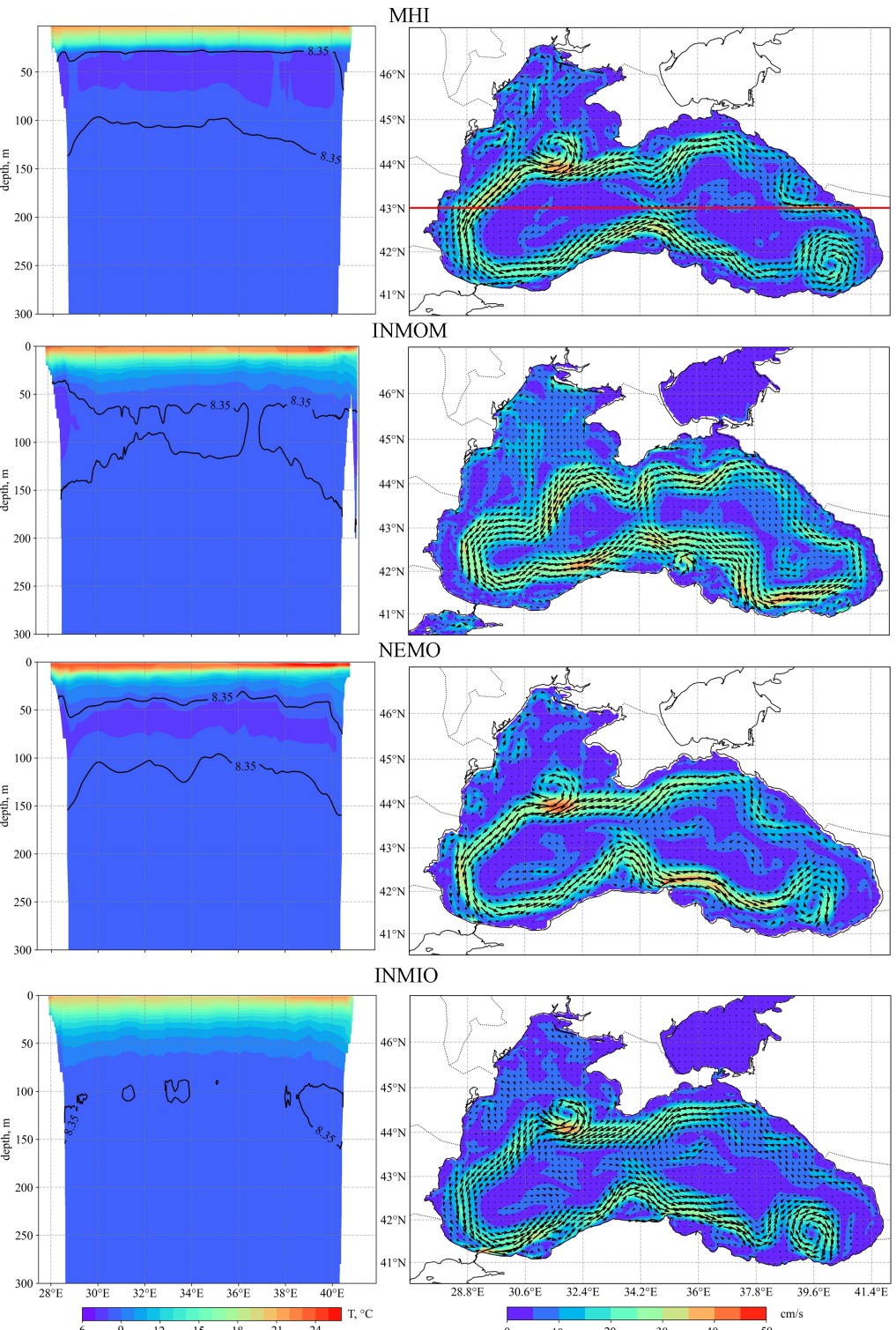

**Figure 2.** Monthly mean temperature in the section along 43° N (left panels) and current velocities at the upper model horizon (right panels) according to the four models for June 2011. Red line is the zonal section along 43° N.

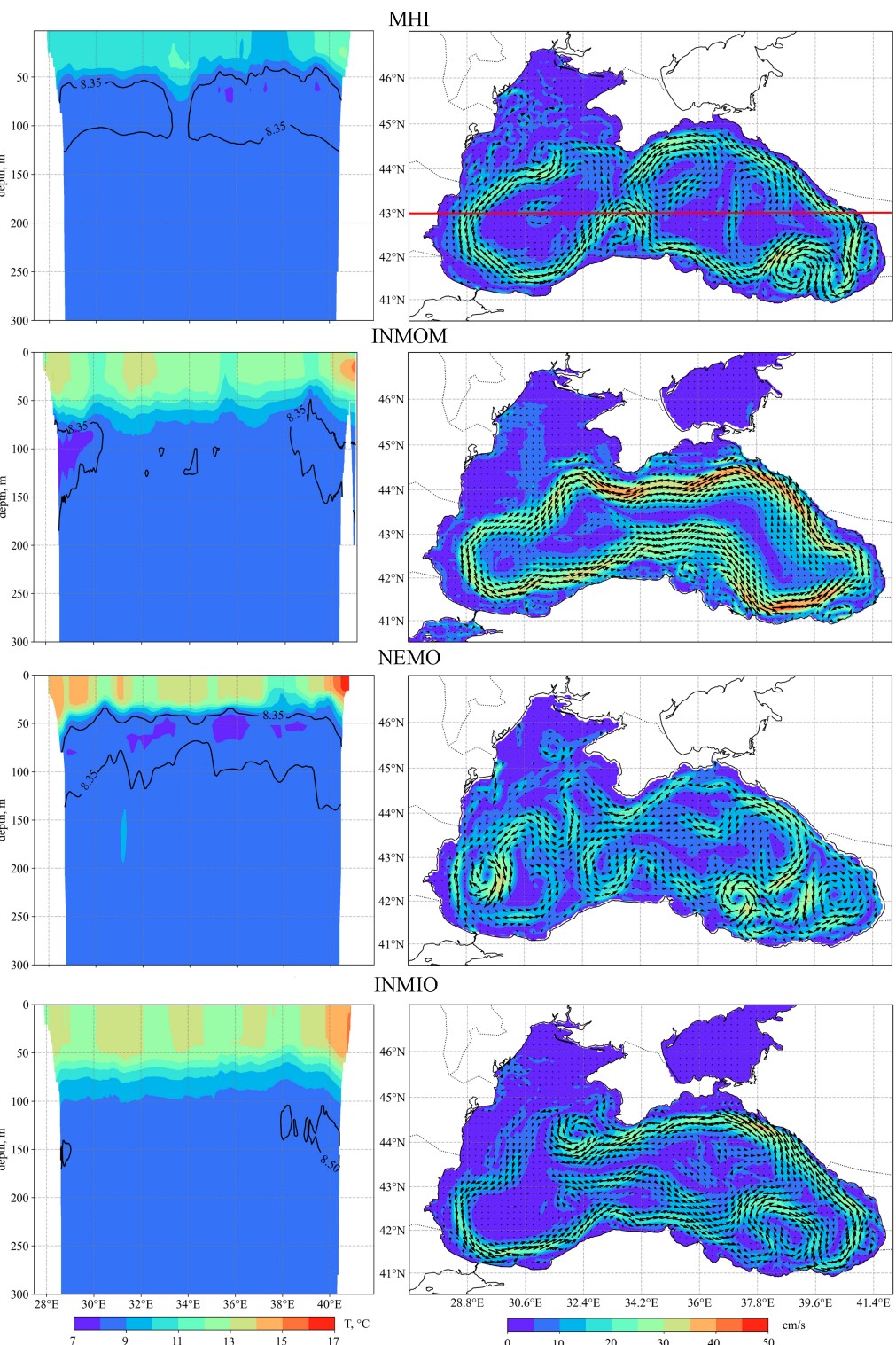

**Figure 3.** The same as Figure 2, but for December 2011.

### 3.3. Deep-Water Circulation off the North Caucasian Coast

For all models, instantaneous fields of current velocity in the summer season show an undercurrent below the main pycnocline in the region of the northeastern continental slope. Thus, in the upper layers, the current velocity vectors are mostly directed to the northwest and correspond to the cyclonic rotation, while the current direction in the deep layers is southeast, i.e., water moves anticyclonically. Figure 4 shows the maps of the velocity field below 900 m near the the North Caucasian continental slope according to the four models.

The most intense anticyclonic currents with a velocity of 5–7 cm/s are found in the MHI simulation (Figure 4a), and the change in the velocity sign occurs at depths below 900 m. According to the other three models, the anticyclonic velocity is 1–2 cm/s. The change in current sign is revealed at depths of about 600–700 m by INMOM and 800–1000 m by NEMO and INMIO.

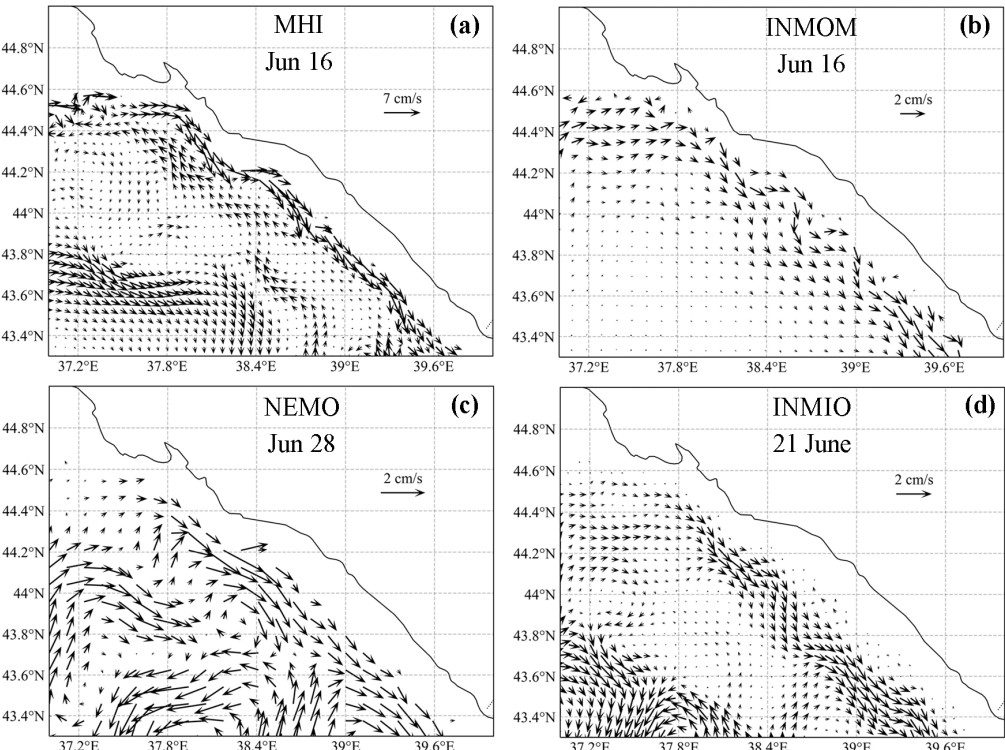

**Figure 4.** Current velocity field in June 2011 according to (**a**) the MHI simulation at the horizon of 1100 m, (**b**) the INMOM simulation at the horizon of 900 m, (**c**) the NEMO simulation at the horizon of 1000 m, and (**d**) the INMIO simulation at the horizon of 900 m.

Analysis of the MHI model velocity fields shows that at a depth of 900–1100 m near the continental slope, intense southeastward currents are observed in all seasons, while in the upper layer, the currents are oriented northwestward [18]. The average velocities of these undercurrents reach 5–7 cm/s during periods from several days to 4–5 weeks (depending on the location and season), the width is 5–8 km, and the length is 200–300 km. It should be noted that the greatest intensity of the undercurrent is reached during the periods of weakening Rim Current, when its average velocity does not exceed 25–30 cm/s. With increasing depth, the magnitude of the undercurrent velocity decreases to 3–4 cm/s.

To analyze the deep-water currents obtained by the INMOM model, the alongshore components of the velocity from June till August 2011 were considered on cross-sections near Bulgarian and Caucasian coasts in [51]. These cross-sections clearly show the presence of an anticyclonic flow at depths of 800 m with velocities reaching 1.5 cm/s (Figure 4b). The velocity of deep-water anticyclonic currents is lower than that obtained by the MHI model on the same cross-sections [52] and measurements. However, these results indicate the presence of certain features of deep-water currents that are not characteristic of the generally accepted scheme of cyclonic water circulation.

The NEMO simulation shows the presence of an anticyclonic current at depths of 800–1500 m. The average velocity in the core of such a flow is about 1 cm/s (Figure 4c), while the average velocity of the Rim Current on the surface is 30–40 cm/s. The lifetime of the anticyclonic current is about 10 days, and it arises periodically at intervals of about a month. To make sure that the detected flow is a current, and not a passing eddy, maps of

the model velocity fields at the horizons of 500, 700, 1000, and 1500 m were studied. It was obtained that the structure is not an eddy, and the jet spreads to significant depths.

According to the INMIO model simulation, a southeastward current with velocities of up to 2–3 cm/s is noted in the velocity field near the North Caucasian coast at depths of 800–900 m from the second half of May till the beginning of September 2011 (Figure 4d). This current is directed against the surface flow, which has a predominantly northwest direction.

The existence of such an undercurrent with the indicated values of the velocity is confirmed by the data of measurements carried out using the moored probe Aqualog in June 2011 [13]. During a full-scale experiment at the Gelendzhik IO RAS test site, a southeastern undercurrent was observed. Its core with velocity of about 3 cm/s is located in the 700–800 m layer, while in the upper layer, the northwestern Rim Current dominates. These data are qualitatively consistent with our numerical experiments, but there are some quantitative differences. When determining the mooring point, it turned out that the model bathymetry does not agree with the real depths. Therefore, a comparison was made for the current velocity according to the in situ data and the MHI and INMOM models at two points: the point (44.47° N, 37.93° E), closest to the measurement site, and the nearby point (44.39° N, 37.86° E), where the model depth exceeds 1000 m. The modeled and measured velocity profiles are presented in Figure 5.

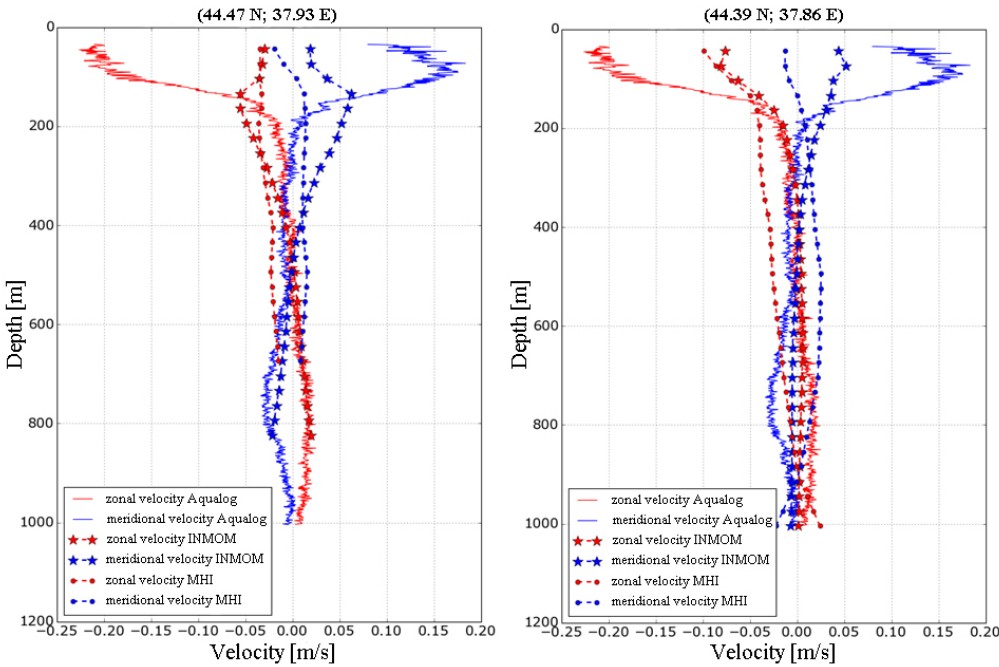

**Figure 5.** Vertical profiles of the zonal and meridional velocity components according to the Aqualog data, the MHI, and INMOM models on June 16, 2011: (**a**) at (44.47° N, 37.93° E), (**b**) at (44.39° N, 37.86° E).

Figure 5 shows that the MHI and INMOM model data are close to the measured values at horizons below 200 m, while in the upper layer, both models underestimate the current velocity. The change of the velocity sign of the measurement data occurs at depths below 500 m. It can be seen that the INMOM model quite accurately reproduces the depth of the sign change at the measurement location and an increase in the velocity at deep-water horizons. The MHI model shows changing in the sign of the velocity starting from a horizon of 900 m and an increase in the velocity comparable with the Aqualog data, but at a deeper horizon (1000 m by MHI vs. 700–800 m by Aqualog). Thus, MHI model velocities weaken with depth more slowly than the measured ones do, and the change in the sign of the modeled velocity occurs deeper. Despite the fact that the velocity increase

according to the MHI and INMOM models occurs at different horizons, the values of both models and measurements of deep currents are close.

Since density currents are most typical in the deep layers of the oceans, horizontal density gradients appear to be the most likely cause of undercurrents in our experiments. We considered the time variation of the velocity and density (salinity) fields of seawater along the continental slope in the northeastern part of the Black Sea. The density in the Black Sea is mainly determined by salinity at depths of more than 50 m [20], so we present here the salinity fields in the region under consideration. A qualitative analysis is carried out using the results of the MHI model. The circulation at depth of 1100 m for different dates is shown in Figure 6, where the salinity field is presented by color (the red line corresponds to the 22.28 ‰ isohaline), and the arrows indicate the velocity field. As seen in Figure 6a, the undercurrent increases in the area southeast of Gelendzhik on 20 June 2011. The cross-shore salinity gradient in this region is higher than, for example, in the area around the point (43.8° N, 39° E), characterized by low current velocity (less than 1 cm/s). Comparing Figure 6a,b, it can be noted that on 28 June 2011, lower salinity is observed near the continental slope, and the 22.28 ‰ isohaline is much closer to the slope. Thus, over time, the salinity gradient increases, and the undercurrent becomes a narrow jet with a velocity of about of 7 cm/s (Figure 6b). After a few days, the cross-shore salinity gradient becomes smaller, and the undercurrent weakens (Figure 6c).

The revealed features of the salinity and velocity field variability correspond to the mechanism of the formation of density currents in the ocean. According to the density currents theory [53,54], the current velocity increases with a growth of the horizontal gradient of seawater density, and in the Northern Hemisphere, the velocity vector deviates to the right from the direction of the density gradient (the gradient is considered to be directed from higher to lower density values). To check the hypothesis about the dominant influence of the density field changes on the formation of undercurrents, we also consider periods when the undercurrent was not observed. Figure 7 shows the salinity and current velocity fields at the 1100 m horizon in the same scales and colors as in Figure 6. It can be seen that the salinity gradients in April (Figure 7a) and November (Figure 7b) are lower than in June (Figure 6), and the undercurrent is not formed. Thus, we suppose that the formation of undercurrents is associated with an increase in the horizontal density gradient, and it is qualitatively confirmed, at least by the MHI model data.

The evidence for the presence of irregular undercurrents offshore the North Caucasus can also be found in the data of ARGO float No. 6901833 (Figure 8) with a parking depth of 200 m. The float trajectories between stations No. 20 and No. 28 performed from 6 September to 15 October 2016 (39 days), and between stations No. 50 and No. 52 performed from 3 February to 13 February 2017 (10 days) show the float's movement to the southeast, in the opposite direction to the overlying Rim Current. The corresponding average velocities of the undercurrents on the 200 m horizon were 2.4 cm/s in autumn 2016 and 8 cm/s in winter 2017.

In addition, by the dynamic (reference level) method [54] using the data of deep-water CTD observations from the R/V Akvanavt at the Black Sea Gelendzhik test site of IO RAS, we estimated the velocities of currents in the northeastern part of the Black Sea. The CTD data obtained in some R/V cruises during the 1997–2008 period have been processed in [21]. The maximum observation depth of 500 m was taken as the reference level. The deep-water undercurrents were detected on some dates. As an example, the coastal component of the current velocity is shown in Figure 9. It can be seen that in late June–early July, the southeastern currents of velocities up to 4 cm/s occur at depths of 200–300 m. In the cross-shore direction, the width of the detected currents is 10–12 km at 200–300 m depth. Due to limited deep-water observation data, it was impossible to determine the along-shore length of the undercurrents found.

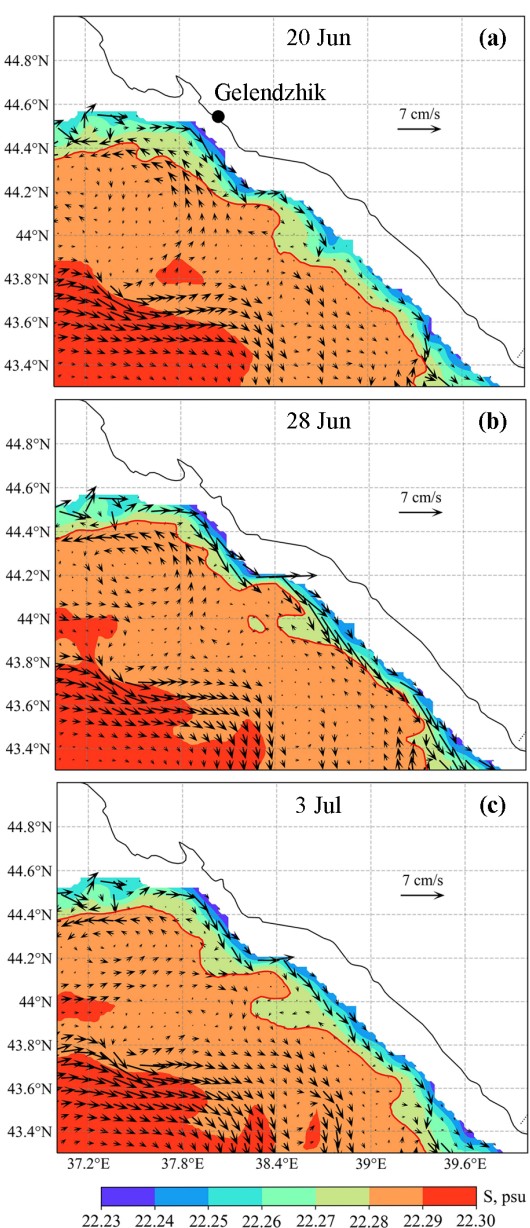

**Figure 6.** Current velocity and salinity fields at the horizon of 1100 m by the MHI model data on (**a**) 20 June, (**b**) 28 June, and (**c**) 3 July 2011. Red line denotes the 22.28‰ isohaline.

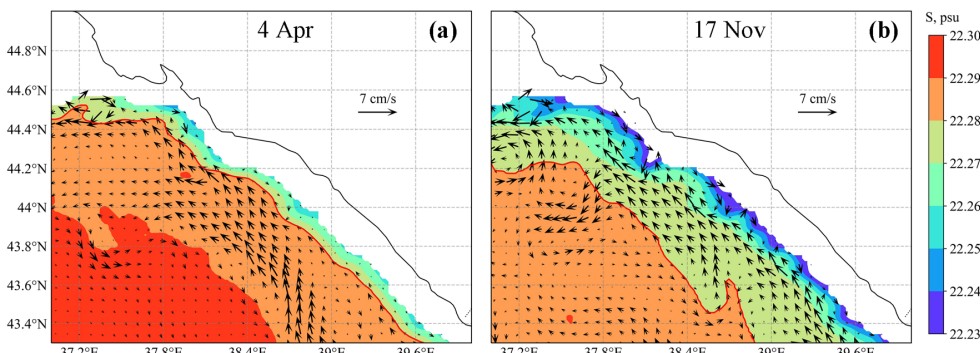

**Figure 7.** The same as Figure 6, but on (**a**) 4 April and (**b**) 17 November 2011.

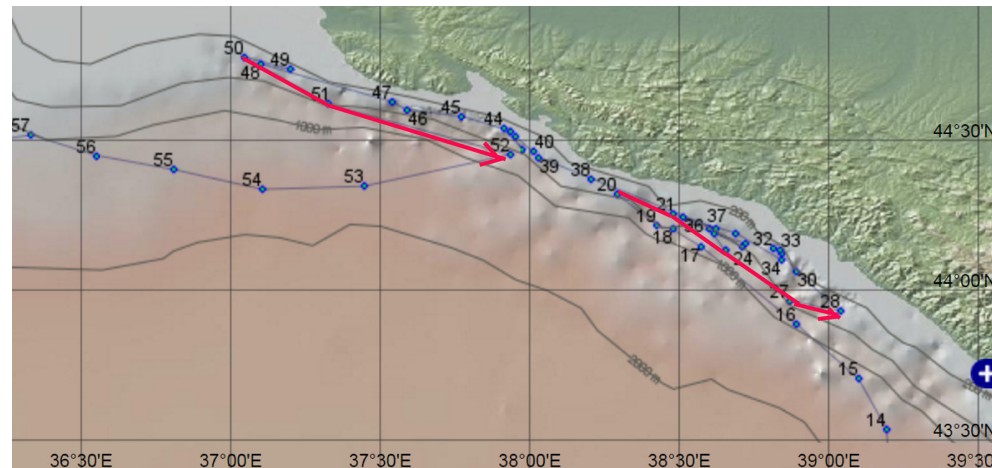

**Figure 8.** The trajectory of ARGO float No. 6901833 (station points with numbers) offshore the North Caucasus coast in the period 07 August 2016–10 March 2017. Red arrows mark undercurrents (http://www.ifremer.fr/co-argoFloats/float?ptfCode=6901833 (accessed on 12 January 2021)).

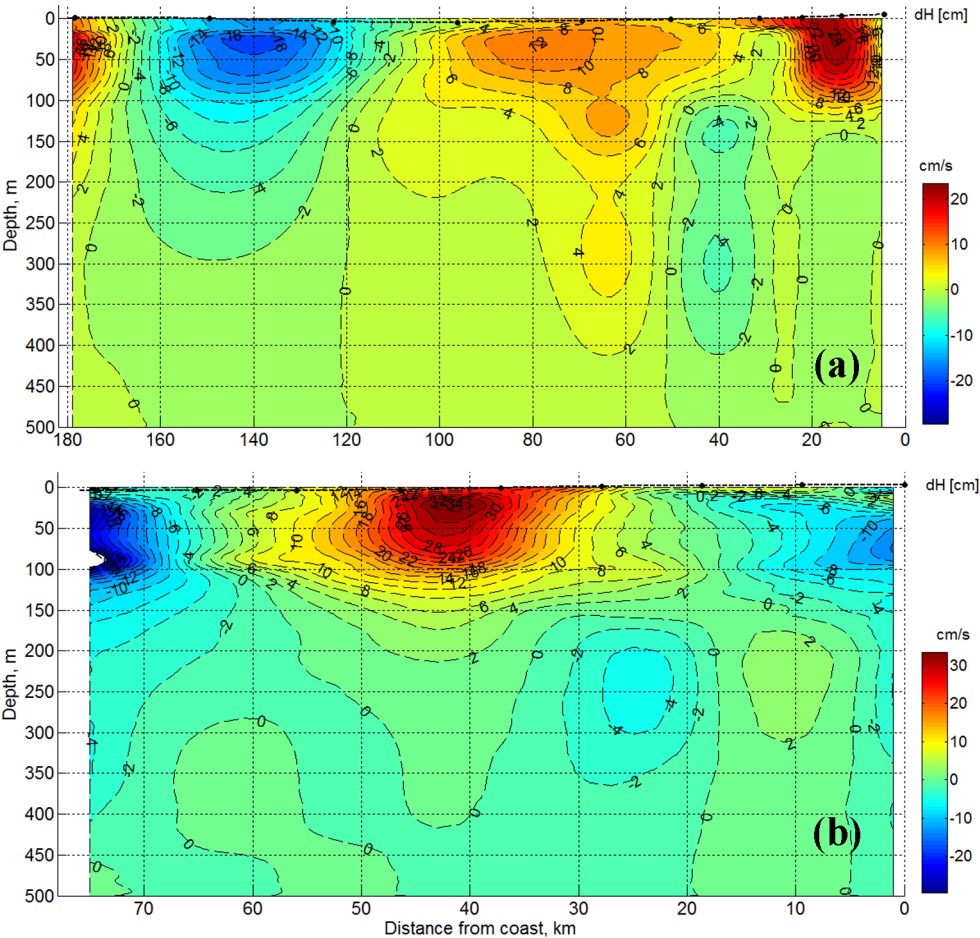

**Figure 9.** Alongshore velocity component near the Black Sea Gelendzhik test site of IO RAS calculated by the dynamic method on the base of R/V Akvanavt CTD measurements in (**a**) July 1998 and (**b**) June 2004. Blue color corresponds to the southeastern current, and dH is the sea surface height.

## 4. Discussion and Conclusions

A comprehensive analysis of the Black Sea hydrophysical fields simulated by four numerical models of ocean dynamics—MHI, INMIO, NEMO (z-coordinate models), and INMOM (σ-model)—is provided for the same one-year period (2011). It is shown that each

of the models realistically reproduces the common features of the basin-scale cyclonic circulation in the upper layer of the Black Sea. Mesoscale variability differs in four simulations and is associated with different turbulent mixing parameterizations and model resolutions. Despite various model settings, all modeling results are generally satisfactorily consistent with T&S profiles observed by ARGO floats in 2011. In the upper 100 m layer, the smallest deviations of T&S are derived from the INMOM model, and below the main pycnocline, they are shown by the NEMO model. At depths of more than 300 m, differences between the modeled T&S fields and ARGO measurement data are quite small.

The features of the current field below the main pycnocline are defined near the North Caucasian coast. Anticyclonic undercurrents spreading along the continental slope are identified in all numerical experiments in summer 2011. The most intense undercurrent with an average velocity of 5–7 cm/s is reproduced by the MHI model, with the change of the current direction from cyclonic to anticyclonic at a depth of about 900 m. According to the INMOM model, the undercurrent velocity is from 1 cm/s up to 2 cm/s, and the change of the velocity sign is at depths of about 600–700 m. According to the NEMO and INMIO models, the undercurrent has approximately the same velocity and is found at depths below 800–1000 m.

A comparison of the simulation results with the in situ data of the current velocity, which were obtained by the Aqualog moored probe during a field experiment in June 2011 at the IO RAS Gelendzhik test site, showed that the highest correspondence between calculated and measured velocities is achieved by the MHI and INMOM models. The values of undercurrent velocities, depth, and life-time are somewhat different depending on the model. These differences are obviously related to the vertical and horizontal resolution of the models used. Thus, the existence of deep-water undercurrents along the northeastern part of the Black Sea continental slope is confirmed by the modeling results as well as by the in situ data. The mean velocities obtained from the results of numerical modeling in the deep-water part of the Black Sea generally correspond to modern concepts. Similar current velocities were calculated using satellite positioning and profiling data of Argo floats in [55,56]. We also found evidence of undercurrents by processing the R/V and ARGO measurements in 1998, 2004, and 2016–2017.

As for the mechanism of the undercurrents' formation near the North Caucasian coast, our preliminary estimates show that the most probable cause of the undercurrents is the horizontal gradient of seawater density in the region. The presence of a density gradient towards the coast forms an anticyclonic current. Further studies may concern the frequency of the occurrence of undercurrents, factors affecting their evolution, and the abundance of such currents throughout the basin.

**Author Contributions:** Conceptualization, S.G.D.; methodology, O.A.D.; software, S.G.D., O.A.D., N.V.M., E.A.K., M.V.S., N.A.T. and K.V.U.; validation, O.A.D. and N.V.M.; formal analysis, O.A.D. and N.V.M.; investigation, O.A.D., E.A.K., M.V.S. and K.V.U.; data curation, N.V.M.; writing—original draft preparation, O.A.D. and N.V.M.; writing—review and editing, K.V.U.; visualization, S.G.D., O.A.D., N.V.M., E.A.K., M.V.S., N.A.T. and K.V.U.; supervision, S.G.D.; project administration, S.G.D.; funding acquisition, S.G.D and O.A.D. All authors have read and agreed to the published version of the manuscript.

**Funding:** This research was funded by the Russian Foundation for Basic Research grant number 18-05-00353 and by the Marine Hydrophysical Institute, Russian Academy of Sciences, according to the state assignment number 0555-2021-0003.

**Data Availability Statement:** The data presented in this study are available upon request from the corresponding author.

**Acknowledgments:** The research was carried out using supercomputer resources at the Marine Hydrophysical Institute (MHI RAS) and the Joint Supercomputer Center of the Russian Academy of Sciences (JSCC RAS).

**Conflicts of Interest:** The authors declare no conflict of interest. The funders had no role in the design of the study; in the collection, analyses, or interpretation of data; in the writing of the manuscript; or in the decision to publish the results.

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
