# Peer review of "Undercurrents in the Northeastern Black Sea Detected on the Basis of Multi-Model Experiments and Observations"

_jmse, doi:10.3390/jmse9090933_

Round 1
Reviewer 1 Report
The paper deals with the important question in the Black Sea physical oceanography on the presence of a deep undercurrent which is below the general cyclonic circulation in the basin. Although some evidences exist on such a current, its characteristics remain unknown. Thus the paper worth to be published in the Journal of Marine Science and Engineering having in mind the aims and the scope of the journal.
General comments:
The paper is interesting, however the authors claim too much, beyond the presented in the paper. The title and the abstract promise more that the paper gives. Based on one year simulation in 2011 the authors claim that the models adequately reconstruct the Black Sea circulation and its seasonal variations. The validation of the models against T&S profiles is not convincing, actually during 2011 little number of Argo floats operated covering mostly the south Black Sea. A figure with the positions of the 360 profiles would be more convincing.
The paper could be published if the authors change the focus of the paper towards the found undercurrent along the North Caucasus coast in the summer of 2011 and investigate the conditions of its appearance during its 10-days lifetime. Could be local wind circulation effects, or coastal waves, or other physical reasons. If the authors investigate the Argo floats trajectories (but not only in 2011) which often pass along the Caucasus coast they could find more evidences on the undercurrent existence. The Argo floats have various parking depth and their displacement is potentially good source of data for the undercurrents.
Some more concrete remarks:
Were the data of MHI Bank used after all? On line 62-68 it is said that the 2011 is chosen because of data availability but is it is not clear what was used.
The model is initialized with TS climate fields on January 2011 but is there a spin-up period? Otherwise, the 1 year simulation could be compromised, as it requires about 3-6 months integration till the model adapts the circulation and thermohaline fields.
Figure 1, 2, 3 the scales and styles are different for the different models and the reader is in difficulty to compare the models
The meridional section in 43N and the zonal section in 39E could be indicated as lines on a map somewhere
Figure 1 and 2: the 4 models present quite different picture of the circulation, in general there is the cyclonic Rim gyre, but the position, the speed, the eddies, are different especially in the eastern basin. However, the authors claim, that the models represent adequately the general circulation and its seasonal variations.
Figure 3: it is not clear whether the shown zonal component is a monthly average or a snapshot at what moment. Overall, it is not convincing for the presence of the undercurrent in the four models.
Figure 4: might be better read with streamlines, now it has to be zoomed many times in order to see the arrows; surface field could be also shown for completeness
Figure 5: the moment in time is not clear; is it the mean August 2011, or a fixed date and hour.
The written in the conclusion in lines 341-348 is merely speculation, as the paper does not deal with the subject.
Reviewer 2 Report
Demyshev et al: Detection of the Black Sea Undercurrents on the Base of Multi-model Experiments and Profiling Data
I carefully examined the manuscript, which is in general well-structured, concise and very informative, and written in excellent language. It is also well in scope of the JMSE journal. The research undoubtedly required high computational power and might have been quite expensive with regard to computational time. Four ocean GCM models were applied to investigate the undercurrents in the Black Sea. The results were validated with the temperature/salinity profiles obtained by the ARGO floats. The models succeeded in reproducing the undercurrents and in this way confirmed anticyclonic circulation under the pycnocline of the Black Sea near the North Caucasian coast during the observed period. The results were successfully compared with the ADCP probe at the test site.
The manuscript is more or less ready to be accepted for publication. I only have two remarks, that could improve the overall quality of the manuscript.
- Authors claim that even small T/s discrepancies in layers under the pycnocline can result in relatively strong currents. In my opinion it should be emphasised (and proved) that despite the discrepancies the main currents are in accordance with measurements. In other words, are the differences small enough to reproduce the undercurrents correctly? What does the difference between ARGO and models do? Figure 5 presents good qualitative agreement and shows patterns of meridional and zonal velocities. Could this figure be supported by a table (similar to Table 1) with relative differences in both velocities? It would certainly tell a lot about quantitative agreement of velocities. I do not insist on model-measurements statistic tests (Kling-Gupta, Nash-Suttcliffe), as they do not do justice to any vectorial quantities, but some kind of quantification should be given in order to support the research results and the conclusions better.
- Figure 4 is too small, it is difficult to distinguish the direction of the currents, and when enlarged, the vectors are blurred.
Round 2
Reviewer 1 Report
The second version of the manuscript is improved, new graphical material is added, and now it is more focused and clear. The authors have addressed my comments and suggestions.
I have one comment on the seasonal variations statements. In order to prove that the seasonal variation of the circulation is adequately represented by the models further analysis is needed, and I would suggest that the authors do not bring attention to it. In the abstract to shorten "It is shown that the models adequately reconstruct the basin-scale Black Sea circulation." And in lines 236-238: "In the current velocity fields, the summer weakening of the Rim Current is observed." Otherwise it is taken for granted that one year (2011) is sufficient to give estimate on the seasonal variability. Alternatively, the authors could give a reference to other study addressing the seasonal variability in the 4 models.
